# Multi-Modal Decentralized Interaction in Multi-Entity Systems

**DOI:** 10.3390/s23063139

**Published:** 2023-03-15

**Authors:** Andrei Olaru, Monica Pricope

**Affiliations:** Department of Computer Science and Engineering, University Politehnica of Bucharest, 313 Splaiul Independentei, 060042 Bucharest, Romania

**Keywords:** multi-agent systems, decentralized interaction

## Abstract

Current multi-agent frameworks usually use centralized, fixed communication infrastructures for the entities that are deployed using them. This decreases the robustness of the system but is less challenging when having to deal with mobile agents that can migrate between nodes. We introduce, in the context of the F_LASH-MAS_ (Fast and Lightweight Agent Shell) multi-entity deployment framework, methods to build decentralized interaction infrastructures which support migrating entities. We discuss the WS-Regions (WebSocket Regions) communication protocol, a proposal for interaction in deployments using multiple communication methods, and a mechanism to facilitate using arbitrary names for entities. The WS-Regions Protocol is compared against Jade (the Java Agent Development Framework), the most popular agent deployment framework, with a favorable trade-off between decentralization and performance.

## 1. Introduction

Software agents are programs with the ability to run autonomously to fulfil their goals. An agent-oriented approach is especially appropriate in the context of heterogeneous, open systems in which agents appear and disappear regularly, and in which the network topology changes often.

Mobile agents have the additional ability to migrate between different nodes in the network. Using mobile agents or not is a choice of paradigm. Mobility is an orthogonal property of agents, independent of other properties. While any distributed application can be implemented by only using static agents, development and deployment can be improved by using agents that are mobile, with the effect of transferring not only data, but also behaviors, through the network, while keeping both the behavior and the data encapsulated in an autonomous entity [1].

There are many uses for mobile agents, among the most recent being the utilization in the Internet of Things (IoT) domain, where mobile agents can move between devices to perform computation locally, avoiding the need to consume bandwidth [2,3,4,5]; can be used for security purposes, detecting attacks or ensuring security [6,7]; or for the management of highly adaptable energy micro-grids [8,9,10].

Let us take a running example to use throughout the paper. A smart building contains several thousands of devices—desktop computers, laptops, smartphones, sensors, actuators, networking devices, printers, and projectors—all connected to the network. In order to be able to perform maintenance tasks on those machines that are managed by the organization, two types of mobile agents can be used. Some agents move through the system on various machines to evaluate the state of the machine, maintain the installed software, and review performance information. When a problem is detected, a specialized agent can be created, containing tools dedicated to that specific problem, and is sent to the machine in question. The advantage of mobile agents here is that they can transport both data and code from one machine to another, and they can use autonomous behaviors to evaluate and decide locally on the actions to take.

Multi-agent system frameworks assist in the development and deployment of multi-agent systems (MAS) by allowing the developer to focus on the design and implementation of the behavior of agents and offers functionality for the discovery of other agents and for delivering messages to other agents [11,12]. Frameworks supporting mobility allow agents to interrupt their execution and migrate to other nodes, where their execution is resumed.

The most popular current agent frameworks (Jade, JIAC, JaCaMo, SPADE [13,14,15,16]) use centralized communication—all messages pass through a central node, which routes messages to their destinations. This method is simple, and it is effective, especially in the context of mobile agents, as all the migrating agents also pass through the same central node en route to their destinations, and the central node is aware of the location of each agent. However, in larger scenarios, the central node becomes a bottleneck and a single point of failure. For communication, Jade supports splitting a large system into several *platforms*. Agents can communicate between platforms; however, they must use their complete names (including the IP address of the platform) and cannot migrate between platforms.

While frameworks generally use agents as the only abstraction available to the developer (for instance, in Jade there are only agents and containers that the developer can create and manage), other abstractions may exist. Among these, are artifacts, as proposed by the Agents and Artifacts (A&A) model [17], agent groups and organizations [18], but also the communication infrastructure itself [19]. Instead of limiting our discussion to agents, let us generalize to *entities*—any persistent unit that can be abstracted in a framework.

Research in decentralized agent systems includes dealing with delivering messages to mobile agents and also with coordination and consensus between agents [20,21,22]. However, they do not deal with creating a framework for *general use*, nor do they deal with practical issues that arise in deployments in real-life networks, such as the lack of routability for nodes in the system.

The main challenges in creating a distributed, decentralized mechanism for interactions between entities deal with entity identifiers and with the current locations of entities—how can an entity be addressed by the same identifier while it is migrating between nodes (machines) and how to localize the destination for a message between entities.

This paper introduces mechanisms for the communication between and migration of entities in a multi-entity system, which is decentralized in the sense that the system can be partitioned into any number of regions which do not depend on one another. The mechanism is interoperable with other communication methods, and it is layered in terms of capabilities to be appropriate for a system with a heterogeneous deployment. We call this the WS-Regions Protocol (*WS* stands *WebSocket*—https://en.wikipedia.org/wiki/WebSocket, accessed on 7 March 2023).

We address applications which need to be decentralized, which use mobile agents, and in which mobile agents need to both send and receive messages from any other entity in the system, regardless of their current location.

We have implemented and validated this mechanism using F_LASH-MAS_, a multi-entity deployment framework developed at our university [23]. F_LASH-MAS_ originally stood for a Flexible and Lightweight Agent Shell and the motto of the project is *Easy for beginners, powerful to experts*. F_LASH-MAS_ is open source, and the code is hosted at https://github.com/andreiolaru-ro/FLASH-MAS, accessed on 7 March 2023. The main goal of the framework is to support maximum flexibility in implementation and interaction, while remaining easy to deploy and efficient in execution.

More concretely, in this paper we present the following:A region-based support infrastructure for decentralized interaction between entities, using modern communication paradigms and supporting frequently migrating mobile entities;A solution for integrating multiple support infrastructures in the same multi-entity system, resulting in a multi-modal framework;A means for entities to be able to use arbitrary names to address each other in the context of a decentralized multi-modal system.

Following the general philosophy of F_LASH-MAS_, we have developed our solution in a modular manner organized in independent modules and layers. The decentralized, region-based support infrastructure is implemented on top of any server–client communication infrastructure. The multi-modal communication infrastructure can work in conjunction with any support infrastructure, and the arbitrary entity naming solution can be used with any decentralized support infrastructure to translate short names into complete names and work across multiple support infrastructures. An architectural perspective of these components is presented in Figure 1.

To evaluate the quality of the solution we propose for decentralized interaction, we perform a performance comparison, both qualitatively and quantitatively, between F_LASH-MAS_ and Jade, the current most popular MAS framework. The results of the comparison were satisfactory and proved that our solution compares well to Jade in the various scenarios, while having the additional advantage of robustness in the case of failure.

After inspecting related work in the next section, we present the main architecture and features of F_LASH-MAS_ in Section 3. Our main contributions are introduced in Section 4, and the validation and analysis of results follow in Section 5. Section 6 draws the conclusions.

## 2. Related Work

Mobile agents are currently used in several application areas, but most of them have characteristics in common, such as being heterogeneous, having a frequently changing topology (some of the devices/hosts are mobile), and suffering from low bandwidth network link between at least some of the devices. The most relevant fields are IoT, vehicular networks, wireless sensor networks, and energy micro-grids.

In wireless sensor networks, mobile agents are used to gather information from WSN nodes. Some works only use mobility along a pre-computed itinerary, with no communication [24]. In the cases where communication is needed, centralized communication methods are used [25,26], many times using Jade—the Java Agent Development Framework [27]. This works for small setups or for when the number of messages is low but is not adequate for city-scale WSNs. Research also exists in the field of distributed computing regarding decentralized messaging [28], however these are made to support only fixed message receivers, which are not able to migrate through the network. This makes the problem that we address specific to the field of mobile multi-agent systems.

In energy micro-grids, agents are used thanks to their autonomous capacity to make decisions in the face of arising issues or system faults [10]. Applications in this domain that use a framework use Jade, however, which is not in itself a robust communication infrastructure.

In the field of IoT, Salah et al. use a microservice-based architecture in which each agent exposes a microservice which responds to HTTP requests [2]. Besides having each agent (or at least each node) deal with the issues of exposing a web server to the Internet, routability means that IPV6 addresses need to be used to identify agents, but these addresses do not remain the same when mobile agents migrate to other hosts, hence being unable to keep an invariable identifier. Indeed, other authors building mobile agent systems rely directly on TCP/IP for communication [3,4,5], making it possible only for mobile agents to contact fixed nodes, but impossible for mobile agents to be contacted from the exterior.

Bosse proposes a framework for the deployment of agents written in JavaScript [29,30], allowing for mobility and interactions with IoT, the cloud, and machine learning services. However, it suffers from limitations in terms of addressing and the limited parallelism of JS, being more adequate for simulations than real-life deployments.

Existing models for message delivery in mobile agent systems have been surveyed and compared previously. Deugo [31] compares several classic delivery models from a theoretical point of view, without actual experiments. Hidayat [32] compares some models from a qualitative point of view of the features offered and challenges solved and proposes a new model. Similarly, Virmani [33] and Rawat et al [20] make qualitative comparisons of the state-of-the-art models at the time.

In the qualitative analyses, the features that are evaluated are generally a subset of the following: solution to the tracking problem (when an agent moves after the message is sent, but before the message reaches it), guaranteed message delivery, support for asynchronous communication, delivery in reasonable time, and transparency of location.

In previous work [34], we have implemented several decentralized message delivery models from the literature and performed a detailed, quantitative comparison of their performance in terms of delivery rate, delivery time, and network load. This comparison has been conducted in a simulated environment with different settings for network and node performance.

Message delivery models for communication in mobile agent systems are generally built around several well-known schemata:The *centralized* solution, in which one server is tracking the whereabouts of all agents and forwards their messages accordingly; this is improved by the *home server scheme* where different servers are assigned to different partitions of the agent set, offering a more balanced solution than centralization [35]; hierarchical solutions, based on *domains* or *regions* also exist [36];*Blackboard* solutions in which agents need to visit or contact the blackboard explicitly to receive their messages [37,38];*Forwarding proxy* solutions in which each host remembers the next location to which an agent migrated, and messages will be forwarded along the path of the agent [39]; the *Shadow Protocol* combines the proxy model with the home server model by using proxies, but agents regularly send updates of their location to their home server [40]; *search-by-path-chase* also adds regions for improved location management [41]; a combination of forwarding proxies and location servers is used by MEFS [42,43].

Distributed decision and control are important aspects in MAS-based control of physical robots [44,45,46]. However, this differs from our work in that, while in robotic systems agents remain static to their machines, in our work, agents move between machines. In the former case, distributed communication does not bring issues in terms of addressing agents because agents usually keep the same identifier (e.g., their IP address).

Regarding robustness and resiliency, there is a significant body of research in multi-agent consensus and protection of multi-agent systems against cyberattacks [47,48,49]. While our goal is to build mechanisms for robustness in MAS, we currently only analyze the possibility of creating tools for a general-purpose distributed MAS framework with no single point of failure. At this point, we do not go into the subject of attacks on such a framework, focusing on the analysis of the performance of the framework during normal operation. An element of future work is to verify how the proposed methods function in the face of attacks.

## 3. F_LASH-MAS_ Architecture

F_LASH-MAS_ is a flexible and lightweight agent deployment framework [23]. It is currently implemented in Java, the same as Jade and many other MAS frameworks. F_LASH-MAS_ models a system as a collection of *entities*—persistent, potentially autonomous processes which offer services, make decisions, perform actions, or help in the interaction with the framework or with the environment. F_LASH-MAS_ defines a basic set of *default* entities: nodes, pylons, agents, artifacts, and shards, the implementation of all of which extends a single Java class—the Entity. However, any instance of any other class-extending Entity can be added to the system at runtime and can interact with any other entities without the need of changing the code of the framework. Some instances of entities inspired by our running example are shown in Figure 2.

Entities in F_LASH-MAS_ exist in the context of one another. There are several standard entities:*Nodes* represent the machines that F_LASH-MAS_ executes on, and any entity runs in the context of the local node;*Support infrastructures* are virtual entities spanning multiple nodes, offering services (such as interaction, discovery, etc.) to other entities;*Pylons* are the materialization of support infrastructures. On every node which exists in the context of a support infrastructure, the infrastructure is represented by a pylon. For entities to use the services offered by the support infrastructure, they must exist in the context of their local pylon;*Agents* are autonomous entities that perceive the environment, make decisions, and perform actions;*Artifacts* are entities that facilitate the interaction between agents and the environment, offer reactive services, or offer support for agent groups, organizations, or other interactions between entities, according to the A&A model [17];*Shards* are sub-agent entities which implement specific functionality or behaviors, existing in the context of agents, nodes, or other entities. Shards implement, for example, functionality for messaging, monitoring, remote control, user interaction, etc.

All entities in F_LASH-MAS_ interact via *waves*. A *wave* has a source, a destination, and any number of other 〈*name, value*〉 arguments. Messages between entities can be transferred as waves, but other types of interaction may exist in the system (in our context, for instance, entities also migrate via waves). Currently, F_LASH-MAS_ offers several types of communication mechanisms (including WebSocket, web services-based, ROS-based, MPI), monitoring and control facilities (including remote monitoring and control via a central web-based interface), and automatic GUI (Graphical User Interface) generation (including the ability to display a graphical interface for an entity on a different node than where the entity currently executes). Thanks to its flexibility, the actual method of communication between entities can be decided at runtime, at the code of entities stays the same regardless of how they communicate.

To ensure flexibility in the interaction method, a support infrastructure is implemented via specific pylons and shards. Shards are driven by events in the entity that contains the shard and can submit events to that entity. While usually shards are part of agents—a specific agent implementation which relies only on shards is also defined and is called a *composite agent*—any entity can use a shard to access its functionality.

Most support infrastructures in F_LASH-MAS_ are implemented in two parts: a node-specific part, materialized as a pylon on each node that is part of the support infrastructure; and an entity-specific part, materialized as a shard that must be added to every entity that exists in the context of the support infrastructure and which needs to access its services. While indeed the shard added to the entity must sometimes be specific to the support infrastructure, F_LASH-MAS_ contains a mechanism which can dynamically load the *appropriate* shard for the support infrastructure without needing to modify the code of the entity. Namely, if an entity needs to use a messaging shard for communication, then when it is added in the context of the pylon offering communication services, it will ask the framework to provide it with the appropriate shard implementation. The interaction between entity and shard is standardized as event-driven, while the interaction between the shard and the pylon can be specific to that support infrastructure.

### 3.1. Challenges and Requirements

Previously, F_LASH-MAS_ supported a variety of communication methods based on client–server or otherwise centralized architectures (WebSocket, RESTful web services, the ROS robot operating system).

The objective that has led to this research was to have a means of interaction which is robust, reliable, and decentralized, such that no specific node is a bottleneck or a single point of failure for the system, all while continuing to support entity mobility.

Of course, centralized interaction brings many advantages that are lost with decentralization. The most important one is the ability of the central node to know on which node an entity is located. In the decentralized case, several challenges arise:[C1]*The infrastructure must know where (to which node) to deliver a message for any entity in the system.*When mobile entities exist in the system, their location can only be determined at runtime; in the decentralized case, there cannot exist a single entity which tracks all mobile entities and can know their location at any time.[C2]*The infrastructure must allow an entity to move to any node in the system.*A large multi-entity system may be formed of several regions employing different communication mechanisms, or using separate servers, central to each region. Migration across regions is in this case more difficult to implement. In Jade, for instance, although messages may be exchanged between different Jade *platforms*, migration across platforms is not possible without the use of third-party plugins.[C3]*The infrastructure must be able to deliver all messages meant for an entity, even if that entity is sometimes in the process of migrating.*Migration is a process that does not happen instantaneously, and in that time, messages may be sent to the migrating entity; these messages must not be discarded, even if the entity is not currently able to receive and process them.[C4]*The infrastructure must mitigate damage when a node is lost and should be able to repair connections affected by losing that node*If any node in the system is lost, the system must be able to recover partially or entirely. In the case of centralized systems, loss of the central node is not recoverable. In the case of region-based systems, loss of the server central to a region usually means losing all the information on that region and orphaning mobile entities originating from that region and which are currently located in other regions.[C5]*(Optional) The entities can be addressed (as message destinations) by an arbitrary name, which is not required to contain routing data for any node (e.g., the IP address of a particular node).*Addressing entities by a simple, arbitrary name is easy for developers in frameworks such as Jade. In decentralized systems (including the case of using multiple, interacting Jade platforms), entities must usually be referred to by an identifier which contains the identifier of the region of that entity.

The Jade framework, currently the most popular MAS deployment framework, as well as other frameworks which it inspired, covers the first three challenges for a single-platform deployment, but our experiments have shown that C3 is not fulfilled for cases when agents migrate very quickly (see Section 5.2). In a multi-platform deployment, it only covers challenges C1 and C3 and needs a plugin provided by a third party to cover challenge C2.

## 4. Multi-Modal Decentralized Interaction between Entities

This section introduces a series of proposals for the improvement of the interaction between entities in a decentralized, heterogeneous system.

The WS-Regions Protocol can be used for routing messages in a decentralized system containing mobile entities;Solutions for heterogeneous, multi-modal systems, in which multiple communication mechanisms are used in different areas of the same multi-entity system;The Arbitrary Entity Naming mechanism allows entities to be addressed using arbitrary identifiers, which do not need to contain the identifier of their home region;Finally, we propose a layer of increased efficiency for message routing, which trades space for increased speed in communication.

### 4.1. The WS-Regions Protocol

We have developed the WS-Regions Protocol, taking inspiration from protocols, such as Home Server, Shadow, and RAMD [35,38,40]. Our initial intention was to use the Shadow Protocol as a decentralized interaction mechanism for F_LASH-MAS_; however, there have been several issues with that approach.

Like other mechanisms for decentralized interaction between mobile agents, Shadow assumes uniform lower-level communication between nodes in that any node can communicate directly with any other. As such, it makes sense for the agent to leave proxies to its previous locations. However, in the modern world, relying on this approach works only for PCs in a local network and, in general, for machines with routable IP addresses but fails when using mobile devices (e.g., smartphones), non-TCP/IP communication (e.g., for Bluetooth devices), or machines behind Network Address Translation gateways. Therefore, in F_LASH-MAS_ we have relied heavily on the WebSocket Protocol [50], which allows full duplex communication between any client and a routable server. However, since communication passes through a server, the method proposed by the Shadow Protocol becomes inefficient—a proxy for an entity does not actually forward messages directly to that entity; instead they go through the region’s WebSocket server anyway. In fact, using WebSocket for communication is more like how region-based protocols work, each region having a server which keeps track of agents created within that region.

As such, we have developed the WS-Regions Protocol, in which the multi-entity system is partitioned into several *regions*, each region containing several *nodes* and a single *WebSocket server*.

#### 4.1.1. Entities

There are several types of entities with specific roles in the WS-Regions Protocol:*Regions* are partitions of the multi-entity system. Each region spans one or multiple physical machines (hence, it spans multiple nodes) and contains a server on one of its nodes—the *region server*; each region has a *region identifier* which is a URI (Uniform Resource Identifier), routable from the region server of any other region;Each *node* in the system contains a *pylon* which belongs to one region;The *WS-Regions pylon* contains a client which connects to the region server at startup;Any *entity* which is *created* within a region has a *name* which is unique to that region and encapsulates a *WS-Regions shard* which interfaces with the local WS-Regions pylon. The region where the entity was created is its *home region*.

Leveraging the fact that the F_LASH-MAS_ architecture is modular and flexible, the WS-Regions Protocol can be deployed on top of any client–server communication protocol, be it WebSocket or something different. It only needs to offer a server entity and a client entity that can be used by the region servers and the WS-Regions pylons, respectively.

Formally, a MAS using the WS-Regions Protocol contains regions, nodes, and entities: MAS=〈R,N,E〉, with R a partition of N and N a partition of E:$$\forall N\in N.\exists R\in R.N\in R\wedge ∄R^{\prime }.\left(R^{\prime }\neq R\wedge N\in R^{\prime }\right)$$
$$\forall E\in E.\exists N\in N.E\in N\wedge ∄N^{\prime }.\left(N^{\prime }\neq N\wedge E\in N^{\prime }\right)$$

That is, any node belongs to one and only one region, and any entity executes on one and only one nodes.

For this formalization sketch, we will use the notation ${MAS}^{t}=R^{t},N^{t},E^{t}$ for the state of the system at time *t* and the relation $\in_{t}$ for a membership relation which is true at time *t*. We will also use the membership relation for the relation between entities and regions, i.e., for E∈E,N∈N,R∈R, we will have E∈R⇔E∈N∧N∈R.

For each physical node there is a Node entity that embodies it and is fixed to that physical node (i.e. cannot migrate): $$\forall N\in N.\exists E_{N}\in E,embodiment\left(N\right)=E_{N},\mathrm{and}\;\forall t.E_{N}\in_{t}N$$

Each entity has a home region, where the entity was created:$$\forall E\in E.\exists R_{E}\in R,home\left(E\right)=R_{E},\mathrm{with}\;E\in_{0}R_{E}$$

Each region is identified by a URI. This makes it easy for regions to address each other. Each entity (including entities that embody nodes) is also identified by a URI. The URI of the entity must contain, as a prefix, the URI of its *home region*, where the entity has been created initially. For instance, an entity Printer308 which has been created in region build.ing/P308 will be identified by the URI build.ing/P308/Printer308. This way, it is easy to route messages to entities based on their identifiers. In the paper, we use simplified names for the sake of readability. A discussion on using simpler identifiers is presented in Section 4.3.

#### 4.1.2. Protocol Sequences

We have illustrated in Table 1 the path taken by a wave in every case of home region/current location for the source and destination entities.

However, a decentralized communication protocol is especially challenging when the entities are migrating. The specific challenge here is how to ensure that messages are not lost. For instance, in the last case in Table 1, suppose that *S* sends a message to *D*, and while the message is in transit, *D* migrates to a region $R_{3}$, different from $R_{1},R_{S},R_{D},$ or $R_{2}$. Migration also takes some time, so entity *D* must suspend processing, package itself, get the package sent to the other node, unpack itself, and resume processing of events. Each of these phases takes some time in which messages may try to reach it. The protocol must ensure the message is buffered and is able to reach region $R_{3}$ and, eventually, *D*. More precisely:An entity’s home server must be aware of the change in region *before* sending messages to a region where the entity is not ready to receive them anymore;An entity must suspend processing *after* receiving confirmation that no other messages will arrive in its current region.

Principle 2 above could be avoided or relaxed if messages arriving in the former region of the entity could be sent by that region to the future region of the entity. However, this means that messages coming from the former region and from the home server would have to be synchronized to arrive to the entity after migration in the correct order (supposing, for instance, that multiple messages were sent by the same entity). Keeping principle 2 above also means that message-chasing problems, which occur when an entity migrates extremely frequently [51], can be solved at the level of the entity’s home server rather than other regions sending messages on the path taken by the entity.

For their home server, mobile entities can have one of three states:home—the entity is located within its home region and has a reference to a WebSocket client connected to the region server;remote—the entity is located within another region than its home region;in-transit—the entity is currently in the process of migration between nodes.

For other region servers, mobile entities can be one of the following:quest—when the entity is in this region, which is not its home region; orin-transit—when the entity is preparing to move away from this region.

To apply the two principles expressed previously, we have implemented the following protocol. Table 2 lists the messages in the protocol; Table 3 presents the pseudocode of the processes that happen in a migrating entity and in the relevant regions; and Figure 3 gives a visual perspective on the migration of an agent. The protocol functions are as follows:An entity *E*, with the home region $R_{E}$, currently on $Node_{1}$ in region $R_{1}$, which intends to migrate to $Node_{2}$ in region $R_{2}$, starts by suspending operations; any interactions beyond this point (both incoming and outgoing) will be buffered and processed after migration is completed;*E* sends to its host server $R_{1}$ a req_leave message announcing the migration; any messages for *E* that do not come from its home server will be sent to the home server, as if *E* had already left;The message is relayed to *E*’s home server $R_{E}$ as a req_buffer message, which starts buffering any messages for *E*; the entity is from this point considered in-transit; then, $R_{E}$ sends a req_accept message to *E* confirming the buffering of messages;Upon receiving the confirmation from $R_{E}$, *E* completely halts, packages its data, and asks $Node_{1}$ to deliver the data to $Node_{2}$;The package follows the direct path $Node_{1}\to R_{1}\to R_{2}\to Node_{2}$, as a wave of type agent_content between node entities;Upon arriving on $Node_{2}$, the package is unpacked by the node and the entity is started;*E* sends a connect message to $R_{2}$, which registers it as a guest entity and sends an agent_update to $R_{E}$; *E* then starts resuming operations and processes any interactions buffered when migration was starting;$R_{E}$ registers *E*’s new location, changes its state to home or remote depending on the current region, and sends all the buffered messages to *E*.

The process is, of course, somewhat simplified when the region migrates from, to, or within its home region.

#### 4.1.3. Entity Mobility

A mobile entity which must be able to migrate between two nodes brings some implementation challenges also from the point of view of internal functionality. Note that entities in F_LASH-MAS_, unlike agents in Jade or in JIAC, have no *pre-established* internal structure. Some standard structures exist, but they are not mandatory.

One of these standard structures most used in F_LASH-MAS_ applications, is the *Composite Agent*, an agent which is constructed exclusively as a set of shards, brought together by a queue of events; shards are able to *post* events in a queue, and an event-processing thread takes each event from the queue and disseminates it to each of the shards in the agent before moving on to the next event (see Figure 4).

A similar structure can be assumed for other entities containing shards or other entities containing sub-entities in that a shard can post events to its parent entity, and the shard can receive events which occur in the parent entity. The structure of the entity, however, is arbitrary, and any entity (including shards) can launch several threads, timers, and so on. This brings challenges when having to suspend the execution of an entity so that it can be resumed later.

F_LASH-MAS_ already offers an implementation for the migration process, which is used primarily by the *Mobile Composite* agent implementation, and that serves as a guideline for any entity that wishes to implement mobility. In the rest of this section, we will refer to the implementation of the Mobile Composite Agent, but the same principles apply to other implementations.

The main challenge when supporting mobility of an entity with an arbitrary implementation is suspending its activity and its interactions without losing events or messages during the suspension period. Considering entities as event-based, we implement a solution to this challenge via four *events*:When the process of migration is triggered by some part of the entity, a before_move event is disseminated to all the shards (or other components) of the entity. This is a signal that a suspension of activity follows, and they should stop or buffer any further interaction. The messaging shard sends the req_leave to the region server. Any messages which the entity still wishes to send or which are received by the messaging shard are now buffered according to the principle of suspending interactions.When the confirmation for migration is received by the messaging shard, it closes all means of communication, and it posts an agent_stop event to the entity, signaling that all activity should stop. When this event is processed by all components of the entity, it is ready to migrate.The entity packages itself and passes the package to its current node, which sends the package to the destination node, which unpacks the entity and starts it.Upon startup, the entity posts an agent_start event to all the shards. This means that they should initialize, but it may be that not all the other shards have been started, and interactions are not yet resumed. The messaging shard posts all buffered received messages and sends all buffered outgoing messages. These events will be processed by the entity after the processing of the agent_start event is completed. It also posts an after_move event, which at the time when it will be disseminated to other shards will mean that all the shards and components have started, that pending interaction has been processed, and that the activity of the entity is fully resumed.

This process ensures that all sub-agent entities are aware of the current state of the entity and, if they abide by the semantics of the events, no interaction is lost. For instance, a shard will know that after the before_move event it can still send messages, but these will not be processed until after the migration, and it will know that any interaction it initiates when processing the agent_stop event may be lost because other shards have already ceased activity.

### 4.2. Multi-Modal Interaction

In a complex, decentralized, multi-entity deployment, it may be that various parts of the deployed system use different communication mechanisms. For instance, in a smart building, there may be local area networks (LANs), Bluetooth-connected sensors and devices, wireless sensor networks using proprietary protocols, and so on. Our goal is to be able to model in F_LASH-MAS_ all the entities in such a deployment, including the support infrastructures themselves, and for entities to be able to interact with each other, via the F_LASH-MAS_ framework, regardless of where in the system the entities are located. The main challenges in this case are the following:Being able to address entities in the entire system via unique identifiers;Being able to deliver messages between entities in different areas of the system;Being able to migrate entities between nodes in different areas of the system.

To ensure that any entity can be addressed via a unique identifier, we adopt the solution of prefixing all identifiers with a URI. This URI may be the address of the central server for a centralized support infrastructure, or it may be the address of a region server in a decentralized support infrastructure such as WS-Regions.

Everything would be quite simple if all entities were identified, in general, by a URI and a name, as in the WS-Regions Protocol; however, this is not always the case. Some support infrastructures may support addressing by simple names. Let us take the following example situation in the context of our running example: in a research laboratory located in room *P308* in the smart building, machines are integrated within F_LASH-MAS_ via a support infrastructure using WebSocket communication; each entity in the laboratory has a unique name, as given by the researchers there. For an experiment, a researcher wishes entity *Sun-seer* to access the value read by a sensor *T* in a micro-meteorological station situated on the roof of the building so that it can predict the weather; there, devices communicate via Bluetooth. Communication between regions in the building is carried out using RESTful web services. Of course, *Sun-seer* may use a specific protocol to read the value from *T*, but integration via F_LASH-MAS_ would mean that it would be sufficient for *Sun-seer* to send a message to *T* as with any other entity, regardless of where *T* is located and how communication is actually performed. Here, there are multiple situations:*T* could be a unique name throughout the building, or it may be addressable as entity *T* within the larger context MSM (the micro meteorological station);*Sun-seer* could be a unique name throughout the building, or it may be addressable as entity *Sun-seer* within the larger context of room *P308*.

We will suppose the second case for each of the two entities. Moreover, in case arbitrary names are not available (see Section 4.3), MSM and *P308* are identified within the building by their URIs build.ing/msm and build.ing/p308. However, entity *Sun-seer* is ignorant of the identifier of the room, and within the room’s support infrastructure, its identifier remains just *Sun-seer*.

To support an addressing system and communication in a multi-modal environment, we introduce *Bridge* entities, relying on previous work that we have conducted in the field [52]. They exist in the context of two or more support infrastructures and are able to route messages among these. They also translate the names of sender or receiver entities, making them usable within the target support infrastructure.

Let us use the same example to illustrate how routing using Bridge entities works (also illustrated in Figure 5):*Sun-seer* sends a message 〈*Sun-seer*, build.ing/MSM/*T*, *get T*〉, where the components of a message are the source, the destination, and the content. It is necessary for the entire identifier of *T* to be known if a directory is not used. The message reaches the WebSocket server in room *P308*;The WebSocket server tries to find the name of the destination entity by iterating through progressively shorter prefixes of the destination name. It has no registrations for build.ing/MSM/T or build.ing/MSM, but an entity (call it *Bridge 1*) has registered to *P308* with the identifier build.ing, as an interface to the rest of the building. In the web services infrastructure of the building, *Bridge 1* registered as build.ing/P308/;*Bridge 1* receives the message and uses the web services support infrastructure to route the message to the meteorological station, but not before prefixing the source of the message with the identifier of *Bridge 1* as a web services end point, transforming the message into 〈build.ing/P308/*Sun-seer*, build.ing/MSM/*T*, *get T*〉;The message is routed to another bridge entity, which had registered as a web services endpoint with the identifier build.ing/MSM;*Bridge 2* relays the message to the other support infrastructure it is registered in (MSM), but it first removes the prefix with which it is registered as a webservice end point, so the message is now 〈build.ing/P308/*Sun-seer*, *T*, *get T*〉;MSM also serves as a gateway for the Bluetooth devices in the meteorological station and knows the identifier *T*, so it sends the message to *T*, which can reply;The reply will follow the same path backwards.

Regarding mobility, again leveraging the principles in F_LASH-MAS_, mobility is implemented regardless of the actual communication protocol, provided that the support infrastructure can buffer messages for migrating entities. As such, as long as the entity itself supports migration and the source and destination node support migration (so as to send and receive/unpack the entity), the transfer between the two nodes is performed as a *wave* between the nodes and is routed in the same manner. Moreover, since the entity can *ask* the local pylon which messaging shard it should load, there is no problem in changing support infrastructures after migration.

### 4.3. Arbitrary Entity Naming

In a centralized multi-entity system, using arbitrary names is easy. By arbitrary names, we mean that there is no requirement for a relation between the name of an entity and its origin, location, etc. In a centralized multi-entity system in which migration is supported, there can exist a centralized directory which holds a correspondence between the names and, for instance, the nodes where the entities are located.

In a decentralized system, a directory would be a single point of failure, and if it were to fail, any further communication would be hindered. However, having multiple directories is difficult in a system with migrating entities because latency in synchronization would mean that immediately after an entity migrates, messages may reach its previous host, and if the entity migrated more frequently, messages may even reach hosts several migrations ago. A solution is to use a home server solution, such as the one in the WS-Regions Protocol. Having a home server for each entity means that entities still need to know which is the home server for an arbitrary named entity, but home servers do not change when an entity migrates. It is then the home server which serves as a directory for all entities originating from that region.

We use the following algorithm for arbitrary entity naming: all entities have a *short name*, which is an arbitrary sequence of characters, and a *home server*, which is identified by a routable URI. We can construct an entity’s *long name* by concatenating the identifier of the home server, a slash, and the short name. The short name must be unique to the home server. For example, entity Printer using home server equipment.build.ing/floor3/P309, would have the long name equipment.build.ing/floor3/P309/Printer. If the short name of the entity contains slashes, then a double slash should be used when assembling it to the URI of the home server, so that the short name can be correctly derived from the long name by other entities.

Different entities situated in different areas of the system may have the same short name, e.g., there may be a Printer on the third floor, but also another Printer entity on the seventh floor. Directories where both entities wish to publish their name will have to accept only one of the registrations.

There are several *directory servers*. Each directory server is itself an entity and exists in the context of at least one support infrastructure. It holds a lookup table with the correspondence between short names and long names. There are several methods in which the lookup table can be populated in the following ways:By synchronizing with other directory servers, whose URIs are given in the configuration.By direct request from an entity that wishes that its name is stored by the directory server. The entity may specify that its name:
–Should be published only in that specific directory server; or–Should be published in that server or in any directory servers it synchronizes with at a given maximum number of hops; or–Should be published in that server or in any other directory servers synchronized with each other in a given domain.Manually by configuring the directory server with a file containing a table.

When an entity sends a message, if the destination is not routable by the support infrastructure and if the support infrastructure is connected to a directory server (or more), the name of the destination will be looked up in the respective servers in order to find its home server.

Optionally, a *local directory* entity may be added to a pylon or to a centralized support infrastructure, which splits the long names that are the sources of incoming messages into short names and home servers and stores these correspondences. The same may be done for the destination of outgoing messages. In the case of incoming messages, a short name will only be saved if it does not clash with an existing identical short name for a different entity.

This method does not necessarily need communication to be conducted via the WS-Regions Protocol. It may be applied regardless of the actual communication method to know what location to send a message to, depending on the arbitrary-named destination for that message. Of course, it is more appropriate for decentralized support infrastructures (such as WS-Regions or web services-based methods) or multi-modal communication.

### 4.4. Increased Communication Efficiency

The routing algorithm discussed in Section 4.1 brings a significant penalty in message delivery time in the case in which messages are sent between entities originating in different regions, and the destination entity is migrating. This is because all messages for an entity must pass through the entity’s home region server. This can be mitigated using the *Short Path* Protocol.

The protocol is active on a per-entity basis. For an entity *E*, the protocol is activated when it sends a request to its home server $R_{E}$ using a short-path_activate message (and later canceled by means of a short-path_deactivate message). Say entity *E* is currently located in a different region $R_{1}$, and entity *A* located in region $R_{A}$ wishes to send a message to *E*. There are two paths which could be taken by the message: the “long path” $A\to R_{A}\to R_{E}\to R_{1}\to E$ and the “short path” $A\to R_{A}\to R_{1}\to E$. However, using the short path would encounter issues when entity *E* would leave region $R_{1}$. The initiation of the *Short Path* Protocol works through the following steps:When *A* sends a message to *E*, $R_{E}$ sends an agent_location message to $R_{A}$ containing the URI of $R_{1}$. $R_{A}$ stores the information that it can use the short path for entity *E* (regardless of what entities in region $R_{A}$ will send messages to *E* from now on).$R_{A}$ sends a long-path-stop message to $R_{1}$ via the long path (via $R_{E}$) and a short-path-begin message directly to $R_{1}$, both containing the name *E*.After $R_{1}$ receives the short-path-begin, it will buffer any messages received via the short path for *E* until the long-path-stop message is received to be sure that messages received via the short path do not get delivered to *E* before messages sent via the long path before the Short Path Protocol is initiated.$R_{1}$ will store the information that region $R_{A}$ is sending messages to entity *E* via the short path.

When entity *E* announces to $R_{1}$ that it is leaving for another region, the protocol must be ended:$R_{1}$ sends a short-path-stop message to all regions that were using the Short Path Protocol with entity *E*.$R_{A}$ cancels the Short Path Protocol for entity *E* and responds with a short-path-end message meaning that this will be the last message sent via the short path. It also sends a long-path-begin to $R_{E}$.$R_{1}$ relays all incoming messages for *E*, including all short-path-end messages, to $R_{E}$.When entity *E* arrives in a new region, $R_{E}$ will send, for each region, first the saved messages received before the short-path-end from that region and then any messages received from that region after the long-path-begin message. This ensures that messages sent to *E* from a region arrive in the same order in which they were sent.

## 5. Discussion and Experimental Results

We have analyzed the trade-offs brought by the solutions presented in Section 4, especially in the WS-Regions Protocol. The goals for this analysis were twofold: first, to verify the validity of the protocol with respect to the challenges in Section 3.1; second, to evaluate how decentralization affects the performance of the system, especially in terms of message delivery times. For the latter of the two goals, we made an in-depth comparison with Jade, which is currently the most popular MAS framework and uses centralized communication. The results were very satisfactory.

### 5.1. Experimental Setup

To perform the tests, we developed two testing scenarios, called “intensive” and “mobility”. In both scenarios, there are four nodes, each located on a different physical machine. There are two regions, each containing two of the four nodes. In each scenario, a number of messages is exchanged very quickly to test how responsive the communication protocol is. Moreover, in the second scenario, an agent moves very quickly between nodes to test how well the protocol handles the message-changing problem.

Scenario “intensive”: There are 16 agents, with 4 agents located on each of the 4 nodes (see Figure 6): A total of 8 pairs of agents exchange 200 messages each. Exchanges happen very quickly: whenever an agent receives a message, it immediately replies. The scenario has two variants—when half of the messages are exchanged across the two regions and when all the messages are exchanged within the same region (see Figure 6a,b, respectively).

Scenario “mobility”: There are four agents—A,B,C,D—with each agent located on one of the four nodes (see Figure 7a). Agent *A* migrates between the nodes very frequently, waiting almost no time at all between two subsequent migrations. Agents B,C, and *D* each exchange messages with *A*, in the same manner as in the “intensive” scenario.

For each scenario, each agent receives a script. In Jade, the scripts were implemented by hand. In F_LASH-MAS_, each agent is provided with a shard that reads the script from a .yaml file containing actions triggered by agent events or delays (see Figure 8). The exact same scenarios were used both with the WS-Regions Protocol and with Jade. For Jade, we used version 4.5.0, the most recent version at the time of writing.

We tested the scenarios using four physical machines connected via a local area network. The machines have varied capabilities—two machine learning servers, one desktop PC, and a low-end laptop. The exact capabilities are not important, as the goals are to check that messages are delivered and to compare the times with the Jade framework, using the exact same setup.

To have a correct test methodology and to have all communication performed simultaneously, the script for each agent was started, with good approximation, at the same time by having an agent send a synchronization message that triggers the beginning of the script after some delay. A local entity is used to *mark* the times when the script starts and when the script is completed. In the analysis, we considered how long it takes for each agent to complete its scenario.

### 5.2. Analysis of Results

We analyzed the results obtained in terms of the distribution of scenario completion times. The completion times for each agent vary greatly, mainly because the machines used in the experiments are quite different. That is why we performed the experiments several times, and we analyzed the distribution and the quartiles for the scenario completion times.

For the “intensive” scenario, the results are shown in Figure 9. We compared the “cross-region” and the “isolated” variant of the scenario running on the WS-Regions Protocol, with the same scenario running on the existing F_LASH-MAS_ centralized WebSocket-based infrastructure and with the same scenario implemented using Jade agents. We observe the following:The times for the cross-region variant of the scenario when using WS-Regions are by far the longest, longer by a factor of about three compared to using Jade. This is because cross-region communication for some agent pairs can take three hops (node—region server—regions server—node).However, in the isolated variant, when only communicating within the same region, the times are significantly shorter when using WS-Regions rather than Jade, by a factor of three on average. This is because in Jade, all messages must pass through the main container, whereas in WS-Regions the regions share the effort of routing messages.The centralized WebSocket infrastructure performs better than Jade, but with comparable results. Just as in the case with Jade, WS-Regions performs significantly worse in the cross-region variant and better in the isolated variant.

For the “mobility” scenario, the results are shown in Figure 7b. Because the number of experiments is large, the figure shows a normal distribution based on the completion times using the two infrastructures. We can see that Jade had significantly better performance than WS-Regions. However, there is an important caveat: when inspecting the console output, we could see that, except for the first message, *all of the message exchanges* happened *after* agent *A* was finished with all migration. In WS-Regions, this does not happen: logs show that almost all the messages are delivered as intended as soon as an agent reaches a node, messages are delivered in the correct order, and in the end all messages reach their destination.

We further evaluated how Jade handles the message-changing problem. We changed the behavior of agent *A* to keep migrating for 20 cycles. What happens is that message delivery fails completely because Jade cannot find the agent and does not buffer the messages for *A*. Hence, very few or no messages were delivered. This happened even if we allowed a delay of up to 50 ms in which *A* would have had time to receive its messages and reply to the other agents. Figure 10 shows the console output for Node 0. In this experiment, only four messages were delivered. In contrast, the WS-Regions Protocol was particularly concerned with the message-changing problem, and all messages were buffered until they were delivered to the destination.

We can see from these results that the F_LASH-MAS_ with the WS-Regions Protocol and the Jade framework are comparable. There is indeed a significant penalty in delivery times when messages move across regions, but these are offset by delivery times for messages exchanged within the same region. Moreover, WS-Regions solves the problem of message chasing much more elegantly (and effectively) than Jade. Of course, decentralization also brings the advantage of robustness in the face of node failure.

We can also discuss here what would be an appropriate size for regions. Region size is a trade-off between efficiency and robustness. When there are many regions, and one region server fails, the other regions are unaffected, save for any agents that were guests on the failed node. However, having too many regions means more communication overhead. The correct balance is, of course, up to the developer of the multi-entity system.

### 5.3. Discussion on Robustness

The goal of having created the WS-Regions Protocol is to ensure that there is no single point of failure in a deployed MAS. We have made some observations on the issues that arise when a node fails, both when using WS-Regions and when using Jade, in Table 4.

The WS-Regions Protocol has obvious advantages with respect to Jade. When a node in a deployment using WS-Regions fails, it never brings down the entire system. Of course, this does not happen in a centralized communication infrastructure, where the central node stores all information about the deployment.

In the case in which the failed node is not a region server, the failure is detected immediately by the region server because the WebSocket connection with the server is broken. The region server contains all the names of the entities which were in execution on the node, so these could be re-created if needed. Messages arriving for the lost entities are buffered, or for guest entities, sent back to their home servers.

In the case in which the failed node is a region server, the failure is detected immediately by other region servers because region servers form a mesh of WebSocket connections. Entities in that region will not be able to interact until the region server is recreated. Messages for entities in the region will be buffered by other regions. The same happens to messages for entities in the region that are currently remote. If the region server can be re-created, the system can continue to function normally. If not, nodes in that region will remain orphans, with entities functioning but unable to interact; if support for changing the names of entities exists, remote entities from the lost region could change their names and be adopted by their current regions.

## 6. Conclusions and Future Work

Our general research goal is to make F_LASH-MAS_ a flexible, modular, easy-to-use framework for multi-agent and multi-entity systems in which the model of the system contains abstractions for all the elements in the deployment, including but not limited to support infrastructures, sub-agent entities, deployment partitioning, and so on.

In this context, we developed WS-Regions as a robust, decentralized communication infrastructure for F_LASH-MAS_ entities, which couples principles from older protocols with the use of modern communication methods. This infrastructure is designed so that it is reliable in terms of message delivery, regardless of how frequently the destination entity migrates.

We compared the performance of the WS-Regions Protocol to the Jade framework, as well as to a centralized WebSocket-based infrastructure, on identical scenarios and observed that: (1) the performance for cross-region communication is of the same order of magnitude with that of Jade, albeit worse; (2) the performance for in-region communication is better than that of Jade; and (3) the reliability with which messages reach a frequently migrating destination is much better than in Jade. Hence, we see the trade-off in performance as worth the gain in robustness and reliability.

From our results and the observations that we made, we can conclude that the WS-Regions Protocol has the advantage that it is more robust, and in some cases, it has better performance than a centralized communication infrastructure. The drawbacks of the protocol are increased complexity and the need for message buffers to keep messages while destination entities are migrating.

Our research continues with improvement of the performance in the WS-Regions Protocol and with the design of large-scale scenarios to test the other proposed methods—an enabler for arbitrary entity names and a method for interoperation between multiple communication infrastructures which is transparent to both the source and the destination of the communication. We will further study the robustness of the WS-Regions Protocol, the capacity to recover entities and messages after a failure, and the capacity to react to partial node failures.

## Figures and Tables

**Figure 1 sensors-23-03139-f001:**
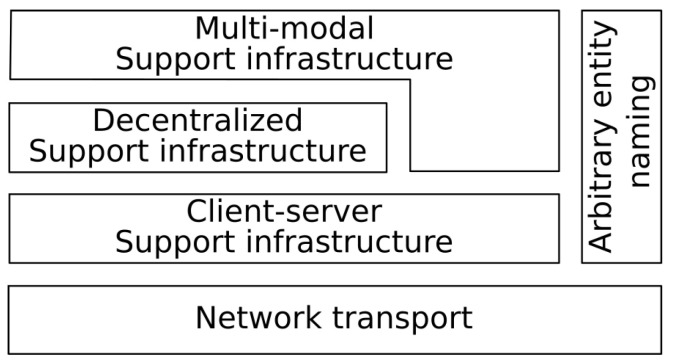
An architectural view of the components presented in this paper.

**Figure 2 sensors-23-03139-f002:**
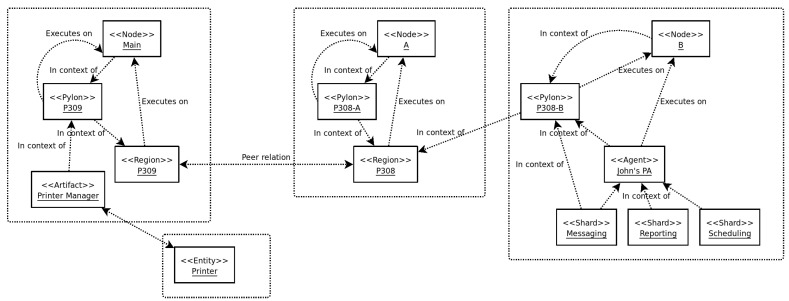
Examples of entities and their interconnections in F_LASH-MAS_ (the Fast and Lightweight Agent Shell). Some relations between entities have been omitted.

**Figure 3 sensors-23-03139-f003:**
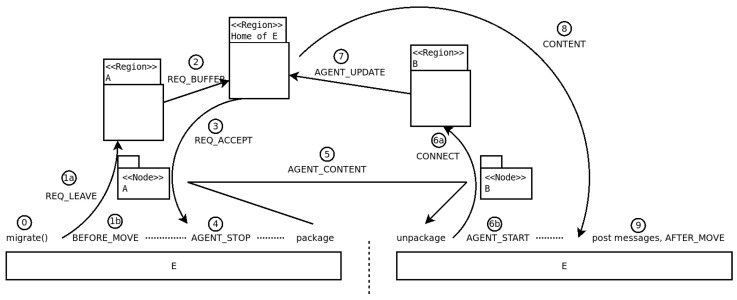
A representation of the process of migration of an entity between two regions of which none is its home region. The phases shown are presented in Section 4.1.2 and Section 4.1.3.

**Figure 4 sensors-23-03139-f004:**
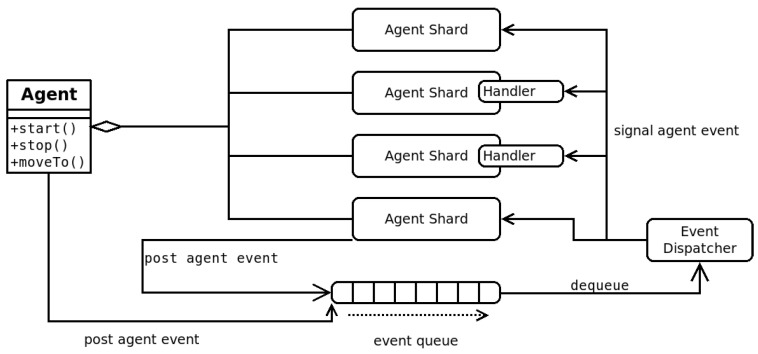
A perspective on how events are processed in a Composite Agent instance.

**Figure 5 sensors-23-03139-f005:**
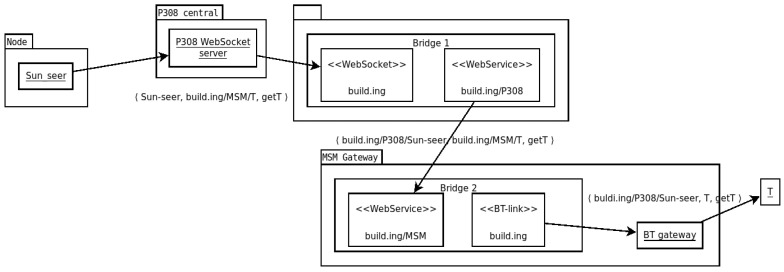
An illustration of the path taken by a message while crossing multiple communication modalities.

**Figure 6 sensors-23-03139-f006:**
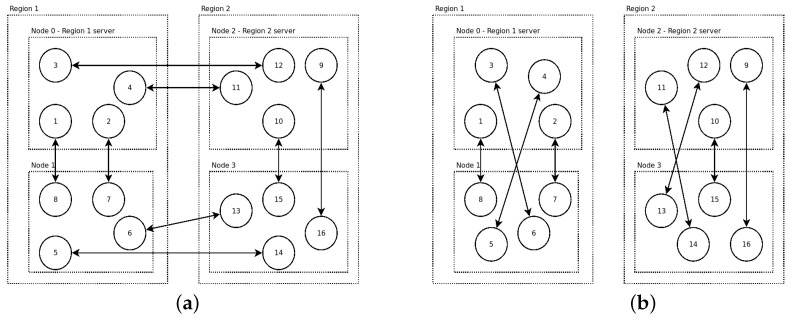
Communication patterns in the “intensive communication” scenario for the normal version (**a**) and the “isolated” version (**b**). Each pair of agents exchanges 200 messages. Note that although the arrows are direct between agents, messages pass through region servers.

**Figure 7 sensors-23-03139-f007:**
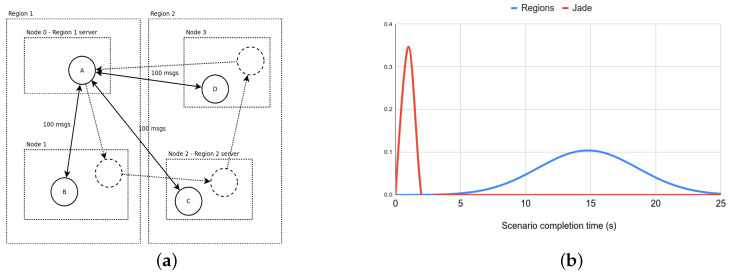
(**a**) The “mobility” scenario. Agent *A* passes 4 times by each of the 4 nodes. Agents B,C,D each exchange 100 messages with *A*; (**b**) the distribution of scenario completion times (for agents B,C,D) when using the WS-Regions Protocol and Jade, respectively. However, there are some caveats.

**Figure 8 sensors-23-03139-f008:**
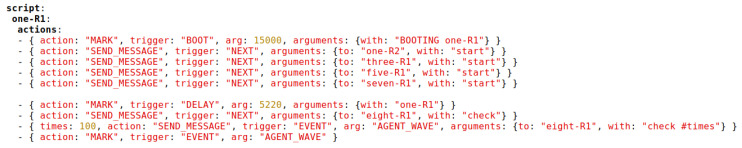
An example yaml script for an agent. Simpler names are used instead of URIs.

**Figure 9 sensors-23-03139-f009:**
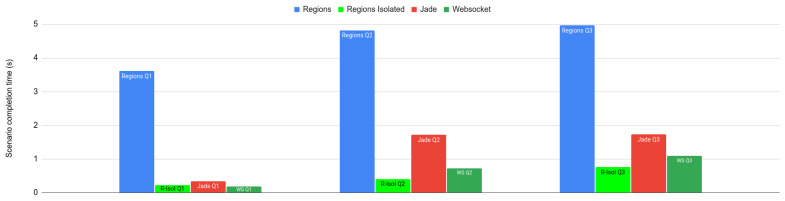
Distribution of scenario completion times (first, second, and third quartiles) for agents in the “intensive communication” scenarios. *Regions*—agents communicate mostly across regions, using the *WS-Regions* Protocol; *Regions Isolated* (*R-Isol*)—agents only communicate within the same region, using the *WS-Regions* Protocol; *Jade*—agents communicate using Jade; *WS*—agents communicate using a centralized WebSocket-based infrastructure.

**Figure 10 sensors-23-03139-f010:**
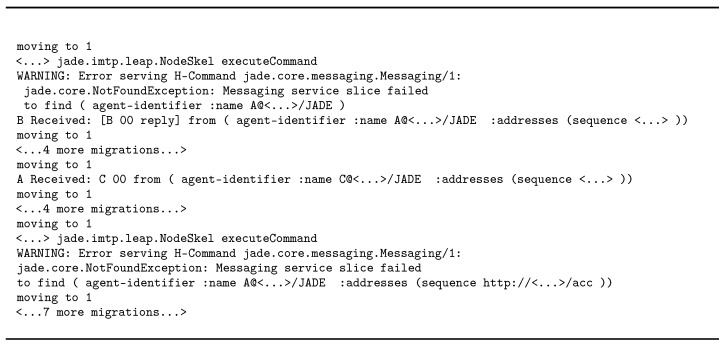
A snippet of the output on Node 0 in the “mobility” scenario, where Jade fails to deliver messages to a fast-migrating agent. Every time the agent is migrating, it is moving to the next node—Node 1.

**Table 1 sensors-23-03139-t001:** The path taken by a wave from entity *S* to entity *D*, considering the same or different home regions, and various situations for their location. In every case, we call the actual nodes where the entities are located $Node_{S}$ and $Node_{D}$, without meaning that those nodes are the same across different cases.

S Home D		S Status	D Status	Wave Path
$R_{SD}$		home	home	$Node_{S}\to R_{SD}\to Node_{D}$
$R_{SD}$		home	remote/*R*2	$Node_{S}\to R_{SD}\to R2\to Node_{D}$
$R_{SD}$		remote/R1	remote/R1	$Node_{S}\to R1\to Node_{D}$
$R_{SD}$		remote/R1	remote/R2	$Node_{S}\to R1\to R_{SD}\to R2\to Node_{D}$
$R_{S}$	$R_{D}$	home	home	$Node_{S}\to R_{S}\to R_{D}\to Node_{D}$
$R_{S}$	$R_{D}$	home	remote/$R_{S}$	$Node_{S}\to R_{S}\to Node_{D}$
$R_{S}$	$R_{D}$	remote/R1	remote/R2	$Node_{S}\to R1\to R_{S}\to R_{D}\to R2\to Node_{D}$

**Table 2 sensors-23-03139-t002:** Types of *waves* sent between entities in the system in the context of the migration of an entity between two host regions, different from its home region.

Message Type	Direction	Semantics
REGISTER	entity → home server	A new entity has been created in the region
CONNECT	entity → host server	The entity has arrived in the region after migration
REQ_LEAVE	entity → host server	The entity wants to move to another region
REQ_BUFFER	host server → home server	An entity in the source region wants to move to another region
REQ_ACCEPT	home server → entity	Acknowledgment of buffer request
AGENT_UPDATE	host server → home server	An entity has arrived in the region
AGENT_CONTENT	node → node	A wave that contains a packaged entity
CONTENT	entity → entity	A wave that is a normal message between entities

**Table 3 sensors-23-03139-t003:** Pseudocode for communication and migration-related processes in the WS-Regions Protocol. The various cases for messages received by region servers are organized in the figure for increased readability of the entire process. The cases in the middle column apply to the case when the current region of an entity is not its home region.

Entity *E*	Current Region of *E*, If Different from $R_{E}$	Home Region $R_{E}$
〈*normal operation*〉		receive normal message *m* for *E*, with $home\left(E\right)=R_{E}$ **case** state(E) of: home: send *m* to node(E) remote on $R_{1}$: send *m* to $R_{1}$ in-transit: add *m* to buf[*E*]
	receive normal message *m* for *E*, with $home\left(E\right)\neq R_{1}$ **if** $E\in R_{1}$ **then** send *m* to node(E) **else** send *m* to home(E)	
〈*E* ∈ *N*_1_ ∈ *R*_1_〉	〈*R*_1_ ≠ *R*_E_〉	
〈*E* *needs to migrate to* *N*_2_ ∈ *R*_2_〉 start buffering in/out operations send req_leave to $R_{1}$ 〈*wait for* req_accept〉 suspend all sub-entities package into *p* request $N_{1}$ to send *p* to $N_{2}$	receive req_leave from *E* with $home\left(E\right)=R_{E}\neq R_{1}$ state(E):=in-transit send req_buffer to $R_{E}$ receive req_accept to *E* from $R_{E}$ send req_accept to $N_{1}$ remove *E* from $R_{1}$	receive req_buffer from $E\in R_{1}$ with $home\left(E\right)=R_{E}$ create buf[*E*] state(E):= in-transit send req_accept to $R_{1}$ receive req_leave from *E* with $home\left(E\right)=R_{E}$ create buf[*E*] state(E):= in-transit send req_accept to *E*
〈*E* ∈ *N*_2_ ∈ *R*_2_〉	〈*R*_2_ ≠ *R*_E_〉	
〈*E* *arrives on* *N*_2_ ∈ *R*_2_〉 unpack all sub-entities send connect to $R_{2}$ run buffered in/out operations resume normal operation	receive connect from *E* with $home\left(E\right)=R_{E}\neq R_{2}$ register that $E\in R_{2}$ state(E):=guest send agent_update to $R_{E}$	receive ag_update from $E\in R_{2}$ with $home\left(E\right)=R_{E}$ state(E):=remote **for each** *m* in buf[*E*] send *m* to *E* delete buf[*E*] receive connect from *E* with $home\left(E\right)=R_{E}$ state(E):=home **for each** *m* in buf[*E*] send *m* to *E* delete buf[*E*]

**Table 4 sensors-23-03139-t004:** A qualitative analysis of the robustness of the WS-Regions Protocol compared to Jade for cases when a node fails completely.

Infrastructure	Case	Entities Lost	Interactions Lost
WS-Regions	node *N* which is not region server fails	entities on node *N* are lost, but their names are known.	messages or migrating entities currently on node *N*.
WS-Regions	node $N_{R}$ containing the server of region *R* fails	entities on node $N_{R}$ are lost; guest entities on $N_{R}$ are lost, but their name is known; all entities $E_{i}\in R,E_{i}\notin N_{R}$ have invalid names until *R* is recreated.	messages and migrating entities in transit in $N_{R}$ are lost; entities in *R* on other nodes than $N_{R}$ cannot interact; entities $E_{i}\in R$ which are remote cannot receive messages and cannot migrate.
Jade	node without main container fails	agents currently on the node are lost.	messages or migrating agents currently on node *N* are lost.
Jade	node with main container fails	all agents are lost.	all interactions are lost.

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
