# Peer review of "Multi-Modal Decentralized Interaction in Multi-Entity Systems"

_sensors, 2023, doi:10.3390/s23063139_

Round 1
Reviewer 1 Report
The authors introduce methods to build decentralized interaction infastructures which support migrating entities in the context of FLASH-MAS multi-entity deployment framework.
The paper is well-written and structure. It is also relevant for the Sensors community.
The experiments showed promising results for the proposed approach. I'm wondering how would the proposal works regarding to failures in the system. Have the authors made experiments considering faults in the MAS? What I'm concerned here is because of the buffers and the amount of messages that could be in there if a fault is not detected (such as the agent crashed during migration).
Author Response
We thank you for the kind comments.
The experiments showed promising results for the proposed approach. I'm wondering how would the proposal works regarding to failures in the system. Have the authors made experiments considering faults in the MAS? What I'm concerned here is because of the buffers and the amount of messages that could be in there if a fault is not detected (such as the agent crashed during migration).
We have introduced a new section – Section 5.3 – dedicated to a discussion regarding faults. We have added to future work elements regarding further experiments with more advanced types of faults. In Section 5.3 we have added a Figure detailing the consequences of nodes failing, both in the WS-Regions protocol and in Jade.
Thank you for the valuable review and for the occasion to significantly improve the paper.
Reviewer 2 Report
This paper presents an extension of the multiagent framework developed by the authors using the decentralization of the location support function for agent mobility. The problem defined in the paper is sharable among the software agent developer communities. However, the solution lacks originality and stringency, so the result is trivial. The authors describe only design philosophy without formalization. Therefore, the paper has to be revised carefully according to the following points and questions:
1. The algorithms or procedures have to be described appropriately, such as pseudo or sample codes.
2. What technology makes decentralized operation possible? Where are the entity names stored? When are they updated? They are too unclear.
3. The experimental results should be described more specifically. Especially in comparison with Jade (lines 664 - 674), the description "message delivery fails completely because Jade cannot find the agent..." is not confident and needs evidence. The message lost depends on how to design the coordination in an agent system and is not seem to happen generally. What system has been used as the Jade system?
4. Is Figure 9(b) real data? It looks like the theoretical distribution function. Please improve the correctness of the description. If the plot is analytically explained, it has to be described formally or mathematically.
5. The paper evaluates the proposal by only comparing it with the Jade framework. However, the authors propose the decentralization of the previous system. The comparison with the previous system would explain the advantage well.
5. Abbreviation rule exists in the Journal. Please check the guidelines.
Author Response
[...] the solution lacks originality and stringency, so the result is trivial. The authors describe only design philosophy without formalization.
The reviewer holds a good point. We have introduced in Section 4.1.1. a formalization sketch and we have used the concepts introduced there in the rest of the paper. We have also clarified possible entity states.
1. The algorithms or procedures have to be described appropriately, such as pseudo or sample codes.
Indeed, pseudo-code was missing and it would have been useful. We have added Figure 3, which details the processes happening in the entities and the region servers.
2. What technology makes decentralized operation possible? Where are the entity names stored? When are they updated? They are too unclear.
We have clarified how entity names are constructed and how they allow routing, by being URIs. Entity names do not change, even when entities are migrating, hence the advantage of the proposed protocol.
3. The experimental results should be described more specifically. Especially in comparison with Jade (lines 664 - 674), the description "message delivery fails completely because Jade cannot find the agent..." is not confident and needs evidence. The message lost depends on how to design the coordination in an agent system and is not seem to happen generally. What system has been used as the Jade system?
We have added details about the Jade version in section 5.1, and we have added clarifications in Section 5.2 about the way in which Jade fails to deliver messages to fast-migrating agents, together with a figure (Fig. 11).
4. Is Figure 9(b) real data? It looks like the theoretical distribution function. Please improve the correctness of the description. If the plot is analytically explained, it has to be described formally or mathematically.
Figure 9(b) is a normal distribution created using real experiment data (which does follow a normal distribution). We have added comments clarifying this in the caption and in the text.
5. The paper evaluates the proposal by only comparing it with the Jade framework. However, the authors propose the decentralization of the previous system. The comparison with the previous system would explain the advantage well.
We have added in the comparison experimental data from running the same scenario using the centralized WebSocket infrastructure. The results are similar with the results obtained with Jade.
5. Abbreviation rule exists in the Journal. Please check the guidelines.
We have made modification so as to abide by the rules in the guidelines.
Thank you for the valuable review and for the occasion to significantly improve the paper.
Reviewer 3 Report
This paper is interesting and well written. There are some problems the authors should consider, which are listed below.
Almost half of the references cited in this paper are quite old. Please update some recent references and make the necessary changes.
The Conclusion section just states what the authors have done. Advantages and drawbacks of the proposed approach should be highlighted, and the future works should be pointed out.
If possible, the authors can provide a comparative study to demonstrate the superiority of the proposed method.
Some latest journal papers on multi-agent systems should be added in the reference in order to give readers a more up-to-date picture. Resilient finite-time distributed event-triggered consensus of multi-agent systems with multiple cyber-attacks.
Author Response
This paper is interesting and well written.
Thank you for the kind comments
Almost half of the references cited in this paper are quite old. Please update some recent references and make the necessary changes.
We have added some newer references and updated the Related Work section.
The Conclusion section just states what the authors have done. Advantages and drawbacks of the proposed approach should be highlighted, and the future works should be pointed out.
We have updated the Conclusion to better focus on advantages and disadvantages. We improved the comments on future work.
If possible, the authors can provide a comparative study to demonstrate the superiority of the proposed method.
We have improved section 5.2 with further experiments, as highlighted in the paper.
Some latest journal papers on multi-agent systems should be added in the reference in order to give readers a more up-to-date picture. Resilient finite-time distributed event-triggered consensus of multi-agent systems with multiple cyber-attacks.
We thank you for the reference, we have analyzed it, together with some related works, and added the comments in the paper.
Thank you for the valuable review and for the occasion to significantly improve the paper.
Round 2
Reviewer 2 Report
The manuscript has been revised sufficiently.
Reviewer 3 Report
No further comments.